# Predicting the Potential Global Distribution of the Plum Fruit Moth *Grapholita funebrana* Treitscheke Using Ensemble Models

**DOI:** 10.3390/insects15090663

**Published:** 2024-08-30

**Authors:** Mingsheng Yang, Yiqi Huo, Lei Wang, Jialu Wang, Shichao Zuo, Chaoyun Pang, Zhengbing Wang, Hongfei Zhang, Kedong Xu, Keshi Ma

**Affiliations:** 1College of Life Science and Agronomy, Zhoukou Normal University, Zhoukou 466001, China; huoyq22@126.com (Y.H.); 15837375363@163.com (L.W.); zy4belorvyilory@163.com (J.W.); 13394849655@163.com (S.Z.); 13468116649@163.com (C.P.); wangzb@zknu.edu.cn (Z.W.); hfzhang_zknu@zknu.edu.cn (H.Z.); 2Field Observation and Research Station of Green Agriculture in Dancheng County, Zhoukou 466001, China; xukd1107@126.com; 3Key Laboratory of Plant Genetics and Molecular Breeding, Zhoukou Normal University, Zhoukou 466001, China; 4Key Laboratory of Crop Molecular Breeding and Bioreactor, Zhoukou Normal University, Zhoukou 466001, China

**Keywords:** Tortricidae, species distribution modeling, climate change, *Grapholita molesta*, pest management

## Abstract

**Simple Summary:**

The plum fruit moth, *Grapholita funebrana* Treitschke (Lepidoptera, Tortricidae), is an important agricultural pest that seriously affects fruit production across the Palearctic region. In this study, for the first time, we predict the potential global distribution of this pest using an ensemble species distribution model. The distribution range predicted, especially for those regions with highly suitable habitats for this moth in parts of East Asia and Europe, indicates a high risk of *G. funebrana* outbreaks and the accompanying massive economic losses, especially in plum and apricot production, highlighting the necessity of pest management. In the United States of America (USA) and Japan (for which *G. funebrana* distributions have not previously been recorded), especially in areas that are highly suitable for this moth, monitoring and quarantine measures should be strengthened to prevent the colonization of this pest and its subsequent wide dispersal.

**Abstract:**

The plum fruit moth, *Grapholita funebrana* Treitschke, is one of the most significant borer pests, often causing huge economic losses in fruit production. However, the potential distribution range of this economically important pest is still poorly understood. For this study, we simulated an ensemble species distribution model to predict the spatiotemporal distribution pattern of *G. funebrana* at a global scale. The results show that the suitable habitats for this moth, under current environmental conditions, are mainly distributed in Europe; East Asia, including China and Japan; Central Asia; and some parts of America. In future projections, the suitable habitats are predicted to generally expand northward, while the suitable area will remain unchanged overall. However, the area of highly suitable habitat will decrease to only 17.49% of that found under current conditions. None of the nine factors used were revealed to be predominant predictors in terms of contributing to the model, suggesting that the integrated effects of these variables shape *G. funebrana*’s distribution. In this study, the distribution range that has been predicted, especially for the regions with a highly suitable habitat, poses a high risk of *G. funebrana* outbreaks, highlighting the urgency of pest management. Moreover, in the United States of America (USA) and Japan (for which *G. funebrana* distributions were not previously recorded), especially in areas highly suitable for this moth, monitoring and quarantine measures should be strengthened to prevent the colonization and further dispersal of this pest, as seen with its close relative *G. molesta*, which has become a cosmopolitan pest species, migrating from its native region (East Asia) to other continents, including the Americas.

## 1. Introduction

The distribution pattern of a species is shaped by multiple factors, especially climate-associated ecological conditions [1]. As the climate changes, a species’ distribution range becomes spatiotemporally dynamic because of alterations in its living environment, including its networks of interaction with other biotic and abiotic factors [2,3]. It is widely acknowledged that understanding the distribution pattern of a species has great implications for further revealing its biogeography, adaptive evolution, speciation, and biodiversity conservation. The known distribution ranges for most indicated species are recorded mainly through information on the available specimen materials examined or via field surveys, while distribution patterns at spatial and temporal scales are generally poorly understood. An important method of effectively revealing spatiotemporal species distribution that is being increasingly widely used is employing species distribution models (SDMs), namely, ecological niche models (ENM). The classical models are usually developed with the following theoretic criteria [4,5]: first, the ecological requirements of a species are acquired through environmental predictor layers linked to the occurrence points; then, those geographic regions meeting the ecological requirements are projected so that they can be defined as a potential distribution range for the examined species. Various methods are used in SDM practice, such as the artificial neural network (ANN) [6], support vector machine (SVM) [7], GAPP [8], and maximum entropy (Maxent) [9] methods, each with their respective advantages and pitfalls. To reduce the impact of individual model biases, an ensemble-modeling method that integrates a suite of different SDMs to estimate habitat suitability via a consensus has been suggested and is increasingly used for various taxon groups, mainly because this method usually provides a more robust methodology than the SDM method alone [10,11,12,13,14,15,16].

The distribution range of a pest species is sensitive to climate and anthropogenic factors, which may affect their population dynamics and can lead to pest outbreaks [1]. In past decades, using SDM methods to predict the distribution dynamics of pests became one of the most important ways of guiding pest management [16,17,18,19,20,21]. The plum fruit moth, *Grapholita funebrana* Treitschke (Lepidoptera, Tortricidae), has a wide distribution encompassing Europe, Asia, and North Africa [22]. *G. funebrana* is oligophagous, feeding on the stone fruits of several hosts, mainly within the plant family Rosaceae [23,24]. As an important agricultural pest, the female adults of *G. funebrana* lay eggs on the exocarp surface of developing fruits [25]. Then, the neonate larvae bore into the fruits, where they feed and develop, leading to yield loss and a decline in fruit quality [26,27]. *G. funebrana* has seriously affected fruit production around the Palearctic region. In Europe, yield losses of up to 40–95% have been reported in plum crops [28,29,30], and a level of 38% damage has been recorded in Romania [31]. In China, the recorded fruit infection rates of plum and apricot orchards have been recorded to be as high as 80% and even 100%, resulting in significant economic losses in fruit production [32,33]. *G. funebrana* is closely related to its congeneric pest *G. molesta* in terms of taxonomy, showing a similar morphology and largely sharing the same host plants, and they can even be attracted to the same female pheromone [23,34]. Notoriously, *G. molesta*, also called the oriental fruit moth, is a globally invasive species that has expanded its distribution range from its native region (East Asia) to other continents, including the Americas, and has become a cosmopolitan pest of stone and pome fruits [35,36,37,38]. Although *G. funebrana* has not been treated as an important invasive pest and is only listed as a quarantine pest in some countries, such as the United States of America (USA), Brazil, and Egypt (corresponding to the Americas, Europe, and Africa [39]), its close similarity to *G. molesta* indicates that *G. funebrana* harbors a great potential to become an invasive species, especially in the context of climate change, which, like *G. molesta*, globally threatens fruit production and food security.

Given its severe impacts on fruit production and its potential invasiveness, in the present study, based on extensive occurrence data gathered from different sources, for the first time, ensemble models were developed to predict the spatiotemporal distribution pattern of *G. funebrana* on a global scale. Our aim was to improve our understanding of the spatiotemporal distribution patterns of this pest, not only in its known distribution range but also in the unrecorded regions that would be suitable for *G. funebrana* in the future, finally providing reference material for the effective surveillance and management of this economically important pest.

## 2. Materials and Methods

### 2.1. Occurrence Records

In the modeling process, the integrality of input occurrence records in the known distribution range of the modeled species has an important effect on the projection results [40,41]. Thus, the occurrence records of *G. funebrana* were extensively gathered from four sources: the global biodiversity information facility database (GBIF, https://www.gbif.org/species/1736528, accessed on 10 August 2023), the BOLD Systems v4 database (BOLD, http://v4.boldsystems.org/index.php/Public_SearchTerms, accessed on 10 August 2023), the Center for Agriculture and Bioscience International (CABI, https://www.cabidigitallibrary.org/doi/10.1079/cabicompendium.29901, accessed on 10 August 2023), and the published literature, obtained through searching the Web of Science (https://www.webofscience.com, accessed on 12 June 2023) and China National Knowledge Infrastructure databases (https://www.cnki.net, accessed on 10 August 2023). In the literature, the distribution records were provided without coordinates but with definite positions that were converted into their coordinate forms on https://api.map.baidu.com/lbsapi/getpoint/ (accessed on, 10 October 2023). In total, 1206 distribution points, i.e., 19 from BOLD, 63 from CABI, 1053 from GBIF, and 71 from the literature, were initially gathered (Figure 1a, Appendix A). After processing using the R package “spThin” [42] to avoid class imbalance and any spatial bias that could cause model overfitting [43,44,45], 871 records (Figure 1b, Appendix A) that assigned, at most, one record to each raster cell of the environmental layers were finally used in the modeling procedure.

### 2.2. Environmental Variables

Two kinds of environmental factors, i.e., climate and elevation, were used in the present study. All 19 bioclimatic variables and 1 elevation variable were downloaded from WorldClim 2.1 (https://www.worldclim.org/, 1 November 2023) [46]. Given the possible existence of collinearity among the variables [47], variable screening was conducted. In this process, a Pearson’s correlation analysis was conducted using the R package “corrplot” [48]. If the absolute value of a correlation coefficient between two variables was >0.8 [40] (Appendix A), one of them was randomly removed in the R procedure. After screening, nine predictors, comprising eight bioclimatic variables and one elevation variable, were used in the modeling procedure (Appendix A).

The near-current bioclimatic layers representing the period from 1970 to 2000 were used in predictions of the current potential distribution. To evaluate the effects of the elevation factor on the predictions, two variable combinations, i.e., only bioclimate variables (BIOs) and bioclimate plus elevation variables (BIOs + elev), were independently employed in the modeling procedure. For future projections, the bioclimatic layers representing the 2021–2040, 2041–2060, and 2061–2080 periods of low (ssp126) and high (ssp585) greenhouse-gas-emission scenarios found in the Coupled Model Intercomparison Project 6 (CMIP6) version were used. To improve projection accuracy, for each period, three different global circulation models (GCMs) were used, i.e., BCC-CSM2-MR, IPSL-CM6A-LR, and MRI-ESM2-0, which represent the different climate sensitivities to future climate change projections [21,49,50]. All environmental layers were developed with a 5′ spatial resolution.

### 2.3. Model Fitting

The ensemble models were developed with the R package “sdm” [51]. Firstly, each of the performances of twelve commonly used individual models included in the package was evaluated, using the occurrence records and environmental layers as input data. The twelve models were a generalized linear model (GLM) [52], generalized additive model (GAM) [53], BIOCLIM [54], multivariate adaptive regression spline (MARS) [55], flexible discriminant analysis (FDA) [56], support vector machine (SVM) [57], random forest (RF) [58], maximum entropy (MaxEnt) [9], Domain [59], classification and regression trees (CARTs) [60], Maxlike [61], and Glmnet [62]. In this analysis, the “gRandom” method of the “sdmData“ function was used to randomly generate 1000 pseudoabsences [63]. Then, 75% of the distribution data was set as training data and the remaining 25% was set as test data, and the maximum number of iterations was set to 5000 [12,13,21]. Finally, for each model with a ten-fold cross-validation approach, the area under the receiver operating characteristic (ROC) curve (AUC) [64] and the true skill statistic (TSS) [65] were calculated. Second, to improve model accuracy, the top five individual models, with an AUC > 0.96 and a TSS > 0.86, were selected and jointly used to simulate the ensemble models. In this analysis, the “ensemble” function of the R package “sdm” [51] was used to combine the output results of the selected five models with a weighted average approach. The “roc” and “rcurve” functions were used to generate the ROC curves and response curves for each variable, respectively. For the future projections under BIOs, the “ensemble” function with a weighted average approach was employed as well.

### 2.4. Model Evaluation and Analyses

In the ensemble modeling, the average AUC and TSS values were calculated to evaluate model performance. The AUC value was generated from 0 to 1; an AUC of 0.7–0.8 was considered acceptable, 0.8–0.9 was considered great, and >0.9 was considered remarkable [66], whereas a value of <0.5 indicated that a model’s performance was no better than random activity. Because of the equal consideration of the sensitivity and specificity of AUC criteria, which may be misleading [67], the TSS value, representing an improved verification index derived from the Kappa coefficient [13], was also considered. This value is generated from −1 to +1, wherein a value approaching 1 indicates a perfect projection, while values of zero or less indicate that a model’s performance is no better than random activity [65,68]. The prediction maps thus generated showed continuous values of habitat suitability. To clearly display and compare the distribution patterns of different projection results, habitat suitability was classified into four classes, i.e., “highly suitable” (0.6–1), “moderately suitable” (0.4–0.6), “poorly suitable” (0.2–0.4), and “unsuitable” (<0.2) [12,69,70]. The R package “ggplot2” [71] was used to visualize the response curves, which show the probability of *G. funebrana* presence as a change in a given predictor. OriginPro version 2021 (OriginLab Corporation, Northampton, MA, USA) was employed to illustrate the percentage contribution of each variable and the values of AUC and TSS generated in the modeling. To display the changes in the distribution patterns between the current and future predictions, the continuous suitability was reclassed into two levels using a threshold value of 0.2; that is, the prediction map was converted into a binary map showing suitability or unsuitability. In this analysis, maps showing stable, expanding, and contracting regions were generated by comparing habitat suitability under current conditions and climate change scenarios, and the corresponding areas were calculated with ArcGIS 10.4 (Esri, Redlands, CA, USA).

## 3. Results

### 3.1. Model Selection and Evaluation

The GAM, MaxEnt, MARS, RF, and SVM models were selected for both the BIOs and BIOs + elev variable combinations to develop the respective ensemble models (Figure 2). The average AUC and TSS values were 0.978 and 0.876, respectively, for the BIOs combination and 0.978 and 0.874 for the BIOs + elev combination, indicating that the model performance for both ensemble models was excellent and the predicted habitat suitability values were reliable. The AUC and TSS values for each model are shown in Figure 3 and Appendix A.

### 3.2. The Current Potential Distributions under the Effects of Two Variable Combinations

Two projections for the current potential distribution of *G. funebrana* were made based on two variable combinations. Under the effect of only bioclimatic variables, the suitable habitats (Figure 4a) were mainly distributed in East Asia, mainly including China and Japan; Central Asia; Europe; and most parts of the USA. When the elevation variable was added to the modeling, the suitable habitat (Figure 4b) that was predicted was generally identical to that under the effect of BIOs in the distribution patterns of all three levels of suitability.

The total area of suitable habitat was 2571 × 10^4^ km^2^ across six regions, including Asia, Europe, Africa, Oceania, and North and South America (Appendix A), with the low-suitability, moderately, and highly suitable areas being 993 × 10^4^ km^2^, 564 × 10^4^ km^2^, and 1041 × 10^4^ km^2^, respectively. Asia had the highest level of suitable habitats (1198 × 10^4^ km^2^), followed by Europe (855 × 10^4^ km^2^) and North America (435 × 10^4^ km^2^), and the other regions had significantly low levels of suitable habitats (<32 × 10^4^ km^2^). The low-suitability and moderately suitable habitats were primarily distributed in Asia and North America, with total proportions of 85% and 82%, respectively. The highly suitable habitats were mainly distributed in Europe, with a proportion of 67%.

### 3.3. Future Potential Distribution and Change Dynamics

Under the effect of BIOs, six prediction maps (Figure 5) were processed from 18 future projections under the effects of variables representing three periods, two greenhouse gas emission scenarios, and three global circulation models.

The suitable areas (Appendix A) were generally comparable with those in the current projection, and a relatively significant expansion occurred in the scenario of ssp585 for the 2070s, with an increase in area of 23.96%. Compared with the current projection, the low-suitability and moderately suitable areas were significantly higher, with average increases of 51.2% and 77.98%, respectively, across all future climate change scenarios. However, the highly suitable areas significantly decreased to only 17.49% of the highly suitable areas under current conditions. In the future, although the highly suitable habitat area is projected to contract, it is always present in Europe, whereas in China and the USA, there are generally no highly suitable habitats present.

The changing dynamics of suitable habitats under the future scenarios of ssp126 or ssp585 for the 2030s, 2050s, and 2070s, compared with those under current conditions, are presented in Figure 6. Overall, the suitable habitats showed a tendency of northward expansion and southward contraction in all future projections. Under the effects of future climate change, the suitable habitats in Asia and North America showed a marked expansion in scenario ssp585 for the 2070s; there was no significant change in Europe, and a significant contraction in Africa, Oceania, and South America (Appendix A).

### 3.4. The Importance of Variables in Modeling

The contributions in terms of percentages for each variable in the two ensemble models are shown in Figure 7. Under the effect of BIOs (Figure 7a), BIO1 was the highest predictor (19.83%) contributing to the projection, followed by BIO5 (17.63%), BIO7 (14.38%), BIO6 (12.5%), and BIO10 (11.58%). BIO4 made the lowest contribution (6.69%). In general, no predominant variables that showed a significantly high contribution to modeling were present. When elevation was added to the predictors, there was a slight change in the order of contributions in terms of size, with BIO7 being the highest-contributing predictor (18.5%), followed by BIO10 (16.03%) and BIO1 (14.77%), and BIO4 remained the lowest (4.89%). The elevation variable made a contribution of only 0.6% to the modeling.

## 4. Discussion

*G. funebrana*, like its close relative *G. molesta*, which is notorious for its global invasiveness, represents one of the most significant borer pests potentially threatening fruit production worldwide. In this study, from the perspective of biogeography, we simulated the ensemble models to predict the worldwide spatiotemporal distribution pattern of *G. funebrana* for the first time. The high values of the evaluation indexes (AUC and TSS) of the models and the general consistency of the distribution ranges defined by current records and our predictions indicate the reliability of the models.

To date, the presence of *G. funebrana* has been recorded across the Palearctic region and in North Africa (Algeria) [22,72]. In Europe, this species has a wide distribution [73] (see also Figure 1); it was first described in 1835 by Treitschke, using type specimens collected from Germany/Czech Republic. Typically, in some other European countries such as Italy and Switzerland, this pest receives intense research attention because of its economic importance [24,26,27,74]. In our predictions, almost all of Europe was a suitable habitat for *G. funebrana*, a result that is consistent with the distribution range according to occurrence records. Moreover, our results show that most of this range is highly suitable for this pest. In China, *G. funebrana* has been recorded in Heilongjiang in northeast China; Ningxia, Gansu, and Xinjiang in northwest China; and Hebei in central China [72]. The prediction results show that the pest’s suitable habitat covered the abovementioned regions or provinces in China. Moreover, some parts of South China, southwest China, and northeast China, such as Liaoning and Sichuan, in which the presence of *G. funebrana* has not been recorded, even showed high suitability in our predictions, indicating that these regions should also be a point of focus for the monitoring and prevention of this pest. In Japan, *G. funebrana* had once been considered to be present, but this finding is now regarded as a misidentification [75]. Our prediction results show that *G. funebrana* is highly suitable for dwelling in almost the entirety of Japan, and it is essential to verify this conclusion through field surveys.

Most members of the Grapholitini tribe, including the *Grapholita* species, are notable pests that can bore into the roots, stems, fruit, seeds, or buds of many economically important plants [22]. At ports of entry to the USA, numerous interceptions of *Grapholita* species have been reported, and no specimens have been identified as *G. funebrana* [76]. Furthermore, Venette et al. [76] performed a risk analysis of its introduction in most states in the USA and noted a low-to-moderate risk of the establishment of *G. funebrana*, based on climate and hosts [23]. Our predictions consistently reveal that in North America, especially in the USA, there is a wide potential distribution range, and the habitats in the mid-eastern US are highly suitable for *G. funebrana*. These findings raise the possibility that once *G. funebrana* colonizes this region, it may quickly adapt to the new environment and experience population expansion, given the availability of suitable habitats and its high dispersal capability throughout its flight period [77]. Another reason for this presumption is that *G. molesta*, which is genetically closely related to and shares similar host plants with *G. funebrana*, has become a globally invasive species that has expanded its distribution range from its native region (East China) to other continents, including the Americas.

Under climate warming conditions, one of the key strategies for species to adapt to the changing environment is range-shifting to track favorable temperatures and/or satisfy moisture requirements [1,14,19]. In most SDM studies, pest species that are subjected to future climate change have been predicted to experience an expansion of suitable habitat, such as the hemipteran *Riptortus pedestris* [21], the dipteran *Aedes aegypti* [78], and the hymenopteran *Tamarixia radiata* [79]. In contrast, an overall contraction of suitable habitats has been reported for some pests, such as the lepidopteran *Spodoptera frugiperda* [80] and the hemipteran *Dalbulus maidis* [81]. For *G. funebrana*, the suitable areas in future scenario ssp126 in all three periods and ssp585 in the 2030s and 2050s are comparable with those under current climate conditions. In contrast, the pest’s suitable habitat significantly expands in ssp585 for the 2070s, although the highly suitable region is remarkably reduced. The different responses of pests to climate warming clearly demonstrate their different ecological demands [21]. Furthermore, the distribution pattern of suitable habitats shows an obvious tendency of northward expansion and southward contraction in future projections, in accordance with a common result of poleward shifts for pests under future climate change conditions [1,82]. Across different regions, future climate change would make the suitable habitats of these pests expand in Asia and North America in scenario ssp585 in the 2070s, with significant contraction in Africa, Oceania, and South America.

It is widely acknowledged that the distribution range of a species is determined by multiple factors. Among these factors, climate-associated temperature and precipitation are often regarded as having a predominant impact on the geographical range shifts of species [14,19,83], which is probably due to these factors’ strong association with energy and water availability [2,21,84]. In our analyses, the eight climate variables selected are all associated with temperature. Interestingly, no variable was revealed as the predominant predictor contributing to the modeling, unlike in some studies wherein one or two predominant contributing factors were revealed (e.g., [21,79]), indicating that these temperature-associated variables have an integrated effect on the distribution of *G. funebrana*. Also, biological investigations into *G. funebrana* have recognized that temperature is an important driving factor affecting the life history and number of generations in different regions [85,86,87], and the adult moths are most active at night when temperatures reach 18–22 °C [23,88]. The elevation factor, often directly acting as a dispersal barrier or indirectly affecting temperature and precipitation conditions, has been widely used in SDM studies (e.g., [20,21,79]). In our analysis, although the distribution pattern in the low-suitability and highly suitable habitats obviously changed when the elevation predictor was included, it made a relatively low contribution (<1%) to modeling, as also revealed by some previous SDM studies on insects [20,21,80]. A possible reason for this phenomenon may be that the influence of elevation on species distribution in the models is modified by climatic factors because of their close associations [89]. From the perspective of variable importance, the requirements for reliably measuring variable importance are likely more stringent than for creating models with high predictive accuracy; accurately measuring variable importance and influence in SDM studies needs further investigation [90].

As an economically important fruit species, *G. funebrana* is receiving intense research attention owing to the economic and social importance of plum production [24]. In the present study, the potential distribution patterns predicted for *G. funebrana* under both current and future climate change scenarios could provide a vital reference for decision-making concerning this pest. For example, some regions that are highly suitable for *G. funebrana*, especially those where stone and pome fruits are cultivated, should be intensely scrutinized to control this pest. Regarding the management strategies, the use of chemical insecticides is currently the primary means of controlling this pest; however, these have adverse effects on agricultural ecosystems and food security [16,87]. In prior studies, egg parasitoids such as *Trichogramma evanescens* and *T. cacaeciae* have been used against *G. funebrana* [91,92], and releasing the latter even brought a 91% reduction in the fruit damage caused by this pest. Recently, Qu et al. [93] optimized the application of *T. dendrolimi* as inundative releases in the large-scale management of *G. funebrana* in orchards. Thus, the *Trichogramma* spp. were regarded as potential biological control agents in *G. funebrana* management [94]. The spatiotemporal distribution patterns of *G. funebrana* can effectively facilitate the application of this environmentally friendly management strategy in terms of the release time and places for release of these parasitoids of *G. funebrana*.

Some regions, such as the vast areas of the northeastern US as well as Japan, in which no *G. funebrana* distributions have been recorded, are highly suitable for the plum fruit moth. Thus, firstly, extensive field surveys (e.g., pheromone trapping) are urgently needed to verify this species’ inexistence in these regions. Then, strict quarantine strategies should be applied, given the possibility that this moth may quickly adapt to the new environment and may experience a rapid distribution expansion like its close relative *G. molesta*. In detail, the government or institutions should strengthen the supervision of fruits imported from *G. funebrana* distribution regions. At the same time, molecular technologies for its detection and monitoring (e.g., DNA barcoding) could provide the timely identification of *G. funebrana* and distinguish its close relatives [95,96].

## Figures and Tables

**Figure 1 insects-15-00663-f001:**
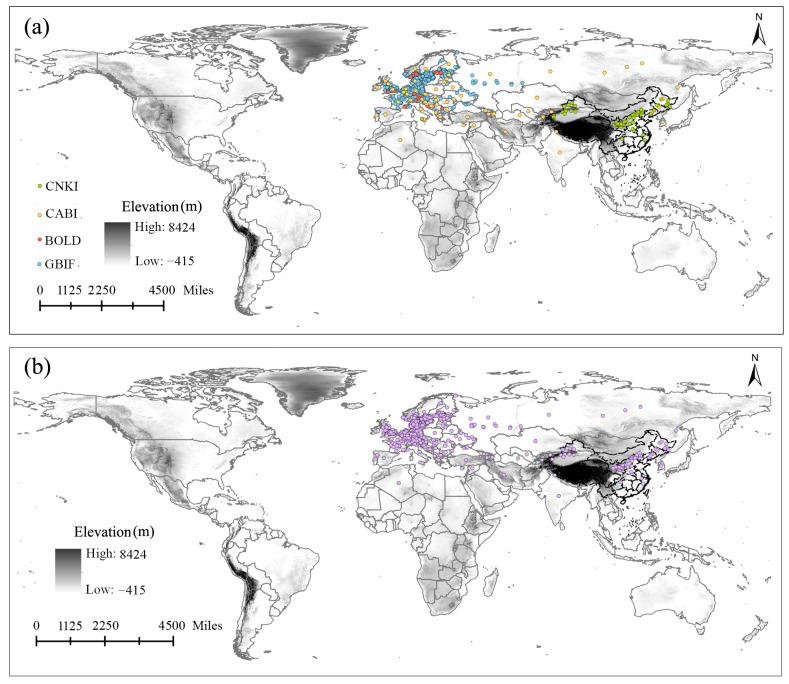
The occurrence data of *Grapholita funebrana* considered in this study. (**a**) The original 1206 occurrence records of *G. funebrana* gathered from various sources. (**b**) The 871 occurrence records used in the models.

**Figure 2 insects-15-00663-f002:**
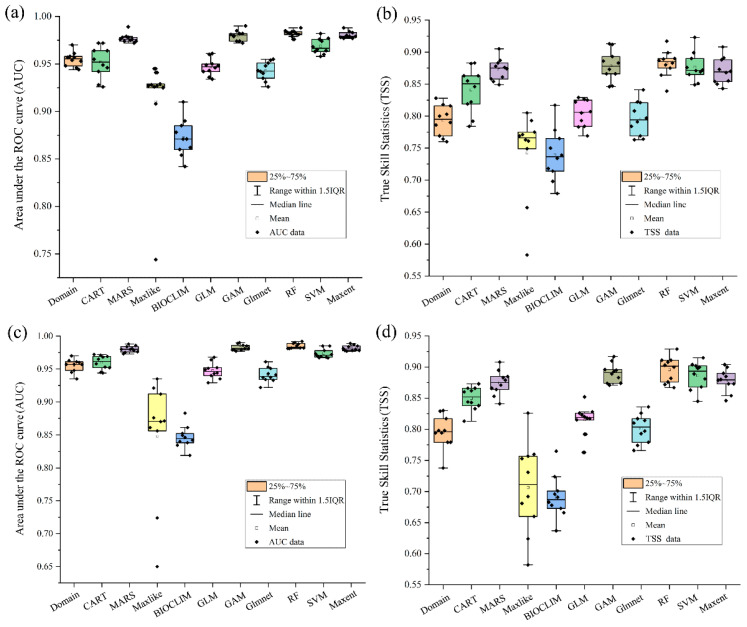
The area under the receiver operating characteristic curve (AUC) and true skill statistics (TSS) values of different single models. (**a**) AUC values under BIOs; (**b**) TSS values under BIOs; (**c**) AUC values under BIOs + elev; (**d**) TSS values under BIOs + elev.

**Figure 3 insects-15-00663-f003:**
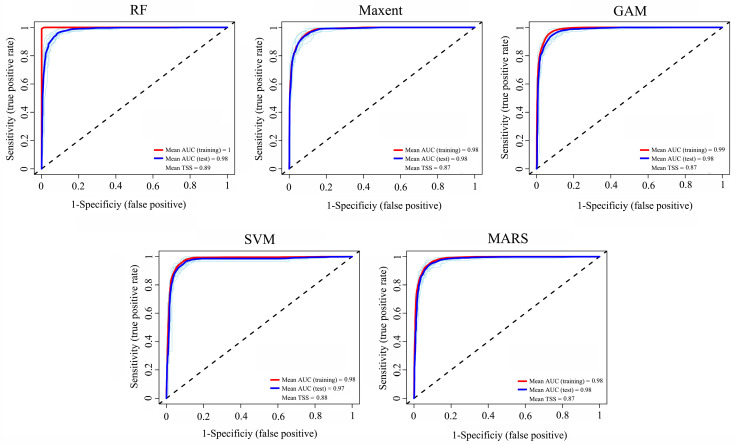
The area under the receiver operating characteristic curve (AUC) and true skill statistics (TSS) values for the five models used under BIOs.

**Figure 4 insects-15-00663-f004:**
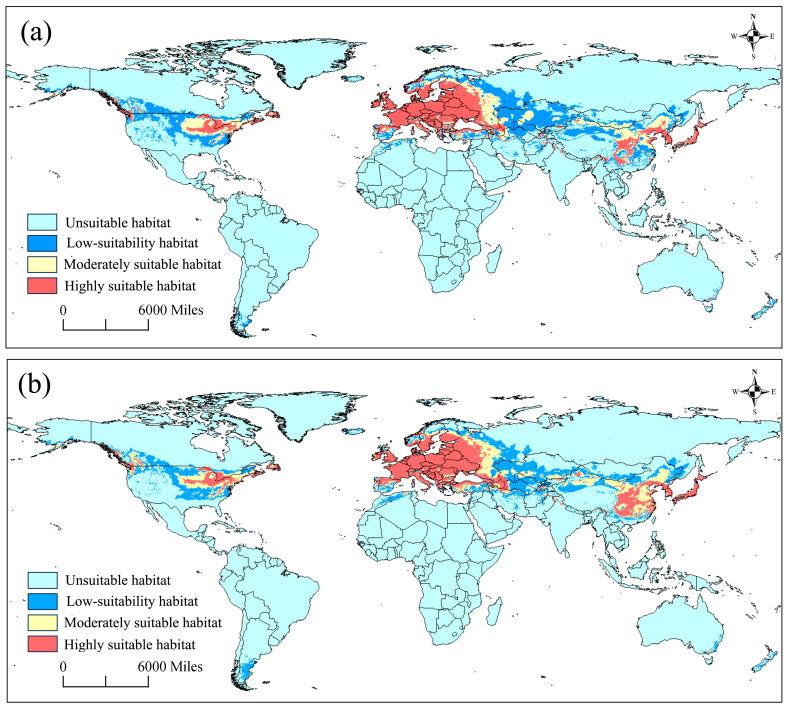
The predicted habitat suitability of *Grapholita funebrana* under current environmental conditions, with two variable combinations. (**a**) BIOs; (**b**) BIOs + elev. 0–0.2: unsuitable, 0.2–0.4: low suitability, 0.4–0.6: moderately suitable, 0.6–1: highly suitable.

**Figure 5 insects-15-00663-f005:**
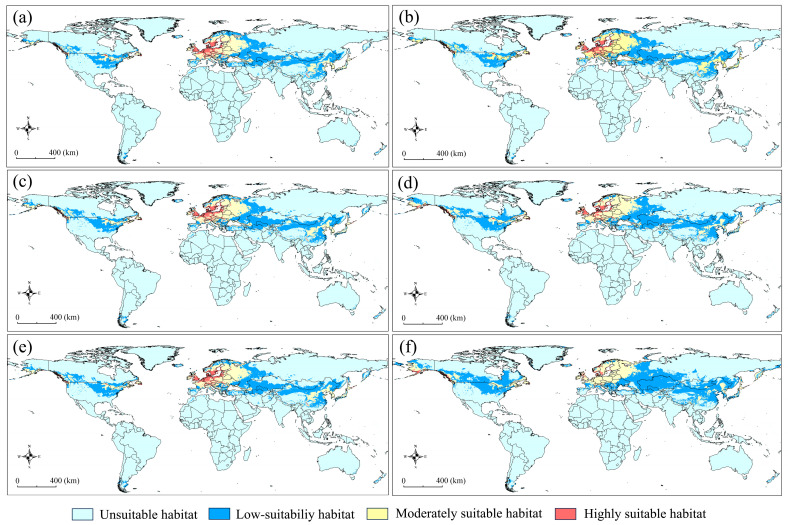
The predicted habitat suitability of *Grapholita funebrana* under future scenarios of climate change. (**a**) 2023s-ssp126; (**b**) 2030s-ssp585; (**c**) 2050s-ssp126; (**d**) 2050s-ssp585; (**e**) 2070s-ssp126; (**f**) 2070s-ssp585. 0–0.2: unsuitable, 0.2–0.4: low suitability, 0.4–0.6: moderately suitable, 0.6–1: highly suitable.

**Figure 6 insects-15-00663-f006:**
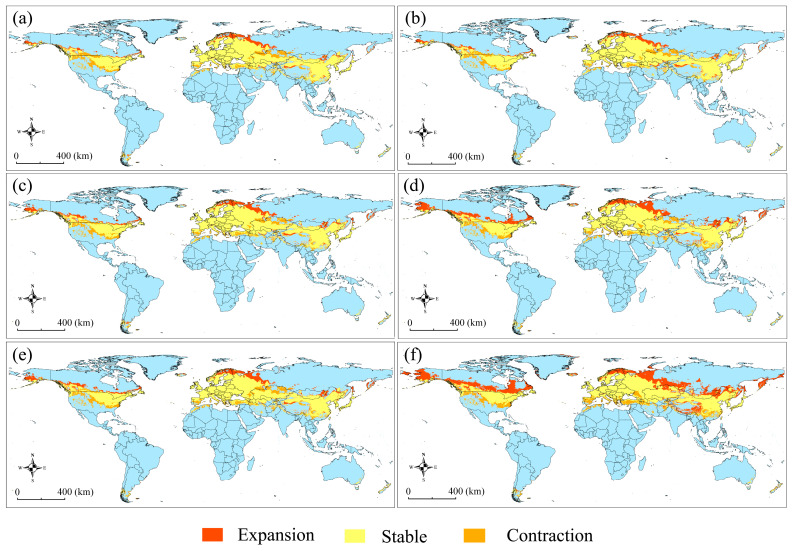
The change dynamics of suitable areas for *Grapholita funebrana* under future scenarios relative to those under current conditions. (**a**) 2023s-ssp126; (**b**) 2030s-ssp585; (**c**) 2050s-ssp126; (**d**) 2050s-ssp585; (**e**) 2070s-ssp126; (**f**) 2070s-ssp585.

**Figure 7 insects-15-00663-f007:**
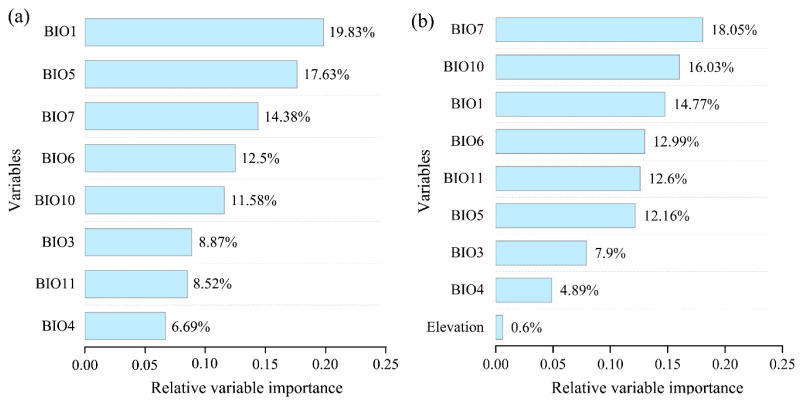
Percentage contribution of environmental variables used in the modeling under two variable combinations: (**a**) BIOs; (**b**) BIOs + elev.

## Data Availability

The original contributions presented in the study are included in the article/Appendix A, further inquiries can be directed to the corresponding authors.

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
