# Peer review of "Predicting the Potential Global Distribution of the Plum Fruit Moth Grapholita funebrana Treitscheke Using Ensemble Models"

_insects, 2024, doi:10.3390/insects15090663_

Round 1

Reviewer 1 Report

Comments and Suggestions for Authors

This seems to be a sound work on modeling of a significant insect pest. Your prediction models indeed show a possible expansion of the past towards the north of the Palearctic, which indeed coincides with several similar SDM's published. What is interesting and baffling at the same time is that your model shows a suitability for the far south of South America in the regions of southern Argentina and Chile. I would sure like you to address this issue as it overall effects the appropriateness of your modeling. It would also be nice to include some more information regarding the natural enemies of this pest and their potential use as biological control agents for pest management. Please give the text to a native speaker so that they can go through several mistakes in grammar and syntax to make your manuscript easier to follow. Finally, it would be interesting to have more information in the Discussion part regarding the actual quarantine and phytosanitary methods used by some of the countries where your model seems to show a high suitability for a new event for invasion.

Comments on the Quality of English Language

The manuscript demonstrates a solid foundation of ideas and research; however, it would significantly benefit from linguistic enhancement. Improving the clarity, coherence, and overall readability will ensure that the content is effectively communicated to the target audience. A thorough review for grammar, syntax, and style will help to refine the manuscript, making it more polished and professional.

Reviewer 2 Report

Comments and Suggestions for Authors While your research topic is of significant interest, there are a number of concerns that need to be addressed before the manuscript could be considered for publication. Firstly, the methodology used in the study lacks sufficient detail and justification. Clarifying the experimental design and statistical analysis would strengthen the credibility of your findings.   Secondly, the results presented in the manuscript are not fully convincing. Some of the key claims lack sufficient evidence to support their validity. We recommend that you conduct additional experiments or analyses to strengthen your conclusions.   Additionally, the writing style and organization of the manuscript need improvement. The introduction section should provide a more comprehensive background and context for your research. The discussion section should critically analyze your results and compare them with previous studies in the field. Comments on the Quality of English Language While your research topic is of significant interest, there are a number of concerns that need to be addressed before the manuscript could be considered for publication. Firstly, the methodology used in the study lacks sufficient detail and justification. Clarifying the experimental design and statistical analysis would strengthen the credibility of your findings.   Secondly, the results presented in the manuscript are not fully convincing. Some of the key claims lack sufficient evidence to support their validity. We recommend that you conduct additional experiments or analyses to strengthen your conclusions.   Additionally, the writing style and organization of the manuscript need improvement. The introduction section should provide a more comprehensive background and context for your research. The discussion section should critically analyze your results and compare them with previous studies in the field.

Reviewer 3 Report

Comments and Suggestions for Authors

Dear authors

I have reviewed your manuscript entitled "Predicting Potential Global Distribution for the Plum Fruit Moth Grapholita funebrana Treitschke Using Ensemble Models." Below, I provide some major suggestions for improving the manuscript. Detailed comments and suggestions can be found in the attached file.

Major Suggestions:

1.      Use of Fixed Thresholds:

Instead of using fixed thresholds for habitat suitability, I recommend using statistically derived best thresholds. This approach can improve the accuracy and reliability of your predictions.

2.      Regional Analysis:

The current global assessment of suitable habitat areas under climate change scenarios overlooks important regional variations. Please include a detailed regional analysis to highlight how different regions are affected differently by climate change.

3.      Averaging AUC and TSS Values:

Averaging the AUC and TSS values of different models may obscure individual model performance. Consider providing the performance metrics for each model individually and explaining the rationale for averaging these values in the context of ensemble modeling.

4.      Clarity and Readability:

Simplify complex sentences and remove redundant phrases throughout the manuscript to enhance clarity and readability. Ensure consistent terminology use (e.g., SDMs vs. ENMs).

Best regards,

Author Response

Reviewer 3

I have reviewed your manuscript entitled "Predicting Potential Global Distribution for the Plum Fruit Moth Grapholita funebrana Treitschke Using Ensemble Models." Below, I provide some major suggestions for improving the manuscript. Detailed comments and suggestions can be found in the attached file.

Major Suggestions:

  1. Use of Fixed Thresholds:

Instead of using fixed thresholds for habitat suitability, I recommend using statistically derived best thresholds. This approach can improve the accuracy and reliability of your predictions.

Re: Thanks for your suggestions and we greatly agreed with your points.

According to your description, the key question should be the selection method of the threshold value used to generate binary distribution maps (presence/absence). In the manuscript, we selected the fixed threshold value 0.2 mainly because of its routine uses in many ensemble modelings in other studies as cited in the manuscript.

According to your suggestions, we tried to calculate the statistically derived threshold value of TSS using the R procedure. In this method, the occurrence data and environmental layers were used as input data, and the main R packages “ENMeval” and “dismo” were used (the R script is attached). By coincidence, the statistically derived threshold value we calculated is 0.1979, which is almost the same as 0.2 we used as fixed threshold value. Thus, in the revised manuscript, the 0.2 is used.

  1. Regional Analysis:

The current global assessment of suitable habitat areas under climate change scenarios overlooks important regional variations. Please include a detailed regional analysis to highlight how different regions are affected differently by climate change.

Re: We followed your suggestions. The suitable habitats were divided into six regions including Asia, Europe, Africa, Oceania, North- and South America, and the habitat areas of all regions, and its effects by future climate change were analyzed in the revised manuscript.

  1. Averaging AUC and TSS Values:

Averaging the AUC and TSS values of different models may obscure individual model performance. Consider providing the performance metrics for each model individually and explaining the rationale for averaging these values in the context of ensemble modeling.

Re: Thanks for your questions.  

In the modeling procedure, generally two steps were needed. Step 1, selecting the top performed individual model from 12 twelve commonly used individual models; according to the AUC>0.96 and TSS>0.86, we selected five individual models to ensure both model diversity and high performances implemented in ensemble modeling. Step 2, the five models were individually re-developed, and the predicted maps of habitat suitability were generated though combining the output results of five models with a weighted average approach. Therefore, averaging the AUC and TSS values is just for easy description and to evaluate the overall performance of ensemble modeling, and the AUC and TSS values for each of the five models are shown in Figure 3.

  1. Clarity and Readability:

Simplify complex sentences and remove redundant phrases throughout the manuscript to enhance clarity and readability. Ensure consistent terminology use (e.g., SDMs vs. ENMs).

Re: We followed your suggestions. The whole manuscript text was revised by the MDPI, and the English-Editing-Certificate is attached. The terminology ecological niche modelings (ENMs) is deleted.

Round 2

Reviewer 2 Report

Comments and Suggestions for Authors

After review, I think the author has carefully and rigorously revised the manuscript. However, I still believe that the language of the manuscript still needs to be checked by professionals.

Comments on the Quality of English Language

After review, I think the author has carefully and rigorously revised the manuscript. However, I still believe that the language of the manuscript still needs to be checked by professionals.

Reviewer 3 Report

Comments and Suggestions for Authors

Dear authors

I am satisfied for your revised manuscript

Well done

With Regards